# Roles of the Oxytocin Receptor (OXTR) in Human Diseases

**DOI:** 10.3390/ijms24043887

**Published:** 2023-02-15

**Authors:** Karolina Pierzynowska, Lidia Gaffke, Magdalena Żabińska, Zuzanna Cyske, Estera Rintz, Karolina Wiśniewska, Magdalena Podlacha, Grzegorz Węgrzyn

**Affiliations:** Department of Molecular Biology, Faculty of Biology, University of Gdansk, Wita Stwosza 59, 80-308 Gdansk, Poland

**Keywords:** oxytocin receptor (OXTR), *OXTR* gene polymorphisms, DNA methylation, gene expression, human diseases, mental disorders, mucopolysaccharidoses, cancer, cardiovascular diseases

## Abstract

The oxytocin receptor (OXTR), encoded by the *OXTR* gene, is responsible for the signal transduction after binding its ligand, oxytocin. Although this signaling is primarily involved in controlling maternal behavior, it was demonstrated that OXTR also plays a role in the development of the nervous system. Therefore, it is not a surprise that both the ligand and the receptor are involved in the modulation of behaviors, especially those related to sexual, social, and stress-induced activities. As in the case of every regulatory system, any disturbances in the structures or functions of oxytocin and OXTR may lead to the development or modulation of various diseases related to the regulated functions, which in this case include either mental problems (autism, depression, schizophrenia, obsessive-compulsive disorders) or those related to the functioning of reproductive organs (endometriosis, uterine adenomyosis, premature birth). Nevertheless, OXTR abnormalities are also connected to other diseases, including cancer, cardiac disorders, osteoporosis, and obesity. Recent reports indicated that the changes in the levels of OXTR and the formation of its aggregates may influence the course of some inherited metabolic diseases, such as mucopolysaccharidoses. In this review, the involvement of OXTR dysfunctions and *OXTR* polymorphisms in the development of different diseases is summarized and discussed. The analysis of published results led us to suggest that changes in *OXTR* expression and OXTR abundance and activity are not specific to individual diseases, but rather they influence processes (mostly related to behavioral changes) that might modulate the course of various disorders. Moreover, a possible explanation of the discrepancies in the published results of effects of the *OXTR* gene polymorphisms and methylation on different diseases is proposed.

## 1. Introduction

Transmembrane receptors, as structures with the ability to receive a specific type of information from the extracellular environment, are key elements in maintaining cellular homeostasis. The signals received by the receptors initiate cascades of signaling reactions, enabling the cell to adapt to the environmental conditions [1].

The oxytocin receptor (OXTR) and oxytocin itself (OXT) are primarily known for controlling maternal behavior. When the fetus develops, OXTR is mainly formed in the myoepithelial cells of the mammary gland and in the myometrium and endometrium of the uterus at the end of pregnancy. OXT is produced by the neurons of the hypothalamus and transported to the higher centers of the brain and the posterior pituitary gland, from where it enters the peripheral circulation. OXT and OXTR take part in the induction of labor (influencing smooth muscle contractions, especially within the reproductive tract) and the initiation and maintenance of lactation, as well as psychological contact between mother and child. They also modulate sexual, social, and stress-related behaviors [2,3]. In addition, they play roles in the development of the nervous system, especially in regulating the growth of the neocortex and maintaining its blood supply, as well as in modulating the autonomic nervous system through the vagal pathway. They also play an anti-inflammatory, antioxidant, and analgesic role, being involved in the prevention of diabetes, dyslipidemia, and atherosclerosis [2].

The molecular mechanism of OXTR action has been reviewed recently [2]; nevertheless, it will be presented briefly here to facilitate understanding and analyzing for further chapters of this paper. OXTR belongs to the G-protein-coupled receptors (GPCRs) family, which is characterized by the presence of the seven-pass transmembrane domain. This transmembrane receptor is coupled with the Gα_q/11_ protein. The signal transduction cascades initiated by the OXTR activation involve several pathways, leading to various effects, including (i) the modulation of the voltage-regulated Ca^2+^ channel and the subsequent activation of the myosin light chain (MLC) kinase, (ii) the stimulation of phospholipase C (PLC) and the activation of protein kinase C (PKC) through the phosphatidylinositol 4,5-bisphosphate (PIP2)–phosphatidylinositol 3 (PI3) pathway, (iii) the activation of the MAP kinase cascade, (iv) cytosolic phospholipase A2 (cPLA2) stimulation, and (v) the activation of the RhoA/Rho-associated protein kinase (ROK) pathway. Therefore, the stimulation of OXTR results in the specific regulation of a battery of genes regulating various cellular and physiological processes. Although early studies reported that OXTR is present mainly in cells occurring in the central nervous system and cells of the mammary gland and uterus during pregnancy [4], subsequent works indicated that this receptor operates also in those building other peripheral organs, such as heart, kidney, pancreas, and thymus [5]. Therefore, the OXT-mediated stimulation of OXTR, and the subsequent activation of various signaling pathways, leading to the specific regulation of expression of many genes, influences various processes crucial to coordinating the physiological processes in response to different conditions, as reviewed recently [2].

The disturbances in the levels of OXT or the activity of the OXTR and their consequences for the functioning of cells and the whole organism have been detected in many human diseases [3]. The most frequently mentioned diseases include those directly related to the central nervous system, such as depression, autism, schizophrenia, or obsessive-compulsive disorders [2], or the functioning of reproductive organs, such as endometriosis and uterine adenomyosis or premature birth [3]. Investigations of changes in the levels or functions of OXT and/or OXTR also apply to cancer, osteoporosis, and obesity, as well as viral infections or cardiac disorders. Those reports presented OXTR as an important factor in maintaining the proper functioning of the body [2,3]. This review not only defines the roles of this receptor in the development of symptoms of various human diseases mentioned above, but also suggests the mechanism of involvement of OXTR in modulating the courses of these disorders.

## 2. OXTR in Reproductive System Diseases

The role of the elements of the OXT system and its receptor is inherently related to the functioning of the endocrine and reproductive system, taking part in maintaining the functionality of the uterus, inducing labor or lactation [2]. It is therefore not surprising that the genetic variability of these elements has been observed in many reproductive system diseases. The role of these factors in the functioning of the reproductive system is emphasized by the mouse model of primary dysmenorrhea, induced by the administration of OXT, which led to pain symptoms caused by abnormal uterine contractions, endometrial edema (edema of endometrium lamina propria), decreased blood flow velocity in the uterine artery, decreased β2-adrenergic receptor levels, and a large increase in OXTR and cyclooxygenase-2 (COX-2) levels in the uterine tissue. These symptoms, especially abnormal contractions and uterine ischemia, reflect the pathology of human dysmenorrhea [6].

DNA sequence analyses of both *OXTR* and the estrogen receptor 1 gene (*ESR1*) in patients with Mayer-Rokitansky-Küster-Hauser syndrome were performed [7]. This condition is characterized by the congenital absence of the uterus and part of the vagina in women. Although disorders of the levels of hormone receptors are mentioned in the literature as one of the elements of pathogenesis, the basic causes of this disease remain unknown. In the study mentioned above, three variants of the *OXTR* gene were detected, of which one (c.-551C > T; rs2301260) was classified as non-pathogenic, the other (c.712G > A; rs61740241) was a missense mutation, and the third (c-133A > C) was of unknown effect on the function of OXTR as this variant was described for the first time. The authors suggested that the identified variants of the *OXTR* gene may disrupt the function of the receptor through various yet unexplored molecular mechanisms [7].

When premenopausal women were screened for *OXTR* expression in two types of affected tissues (peritoneal endometriosis, containing glandular and epithelial cells and ovarian endometriotic cysts), immunohistochemical staining showed very high levels of OXTR both in the cytoplasm and nuclei in the epithelial cells of the functional endometrial layer, and in the cytoplasm of epithelial cells of peritoneal endometriotic lesions [6]. An even higher abundance of nuclear OXTR was noted in endometriotic ovarian cysts. In addition, high levels of OXTR were demonstrated in the smooth muscle cells surrounding the endometriotic lesion. Moreover, OXTR immunofluorescence staining was also performed in an endometriosis cell line, indicating numerous spots irregularly dispersed in the cytoplasm [8]. These early studies clearly showed that the expression of *OXTR* may be enhanced under specific conditions, possibly contributing to pathomechanisms of these reproductive system diseases. Subsequently, a similar study was conducted using tissues with adenomyotic lesions derived from premenopausal women, indicating not only the overexpression of *OXTR* in adenomyosis-associated myometrium, compared to unaffected myometrium, but also significant morphological changes [9]. Therefore, the latter results corroborated the conclusion about the importance of OXTR levels in the pathologically changed tissues.

OXT is critical in inducing and sustaining labor [3]. Therefore, the roles of both OXT and OXTR have been studied in premature births. However, the *OXTR* gene polymorphism studies did not indicate a direct correlation between the presence of any individual polymorphism and preterm birth [10]. Nevertheless, it was found that the haplotype combination of the rs2254298 A, rs2228485 C and rs237911 G alleles was significantly associated with an increased risk of this condition [10]. Interestingly, quite a similar study demonstrated that most of 14 identified *OXTR* variants were not correlated with the risk of preterm birth [11]. On the other hand, two maternal *OXTR* polymorphisms (rs4686302 and rs237902) have been identified to likely contribute to gestational age-dependent effects on prematurity. Moreover, significant differences in the bindings of the ligand by wild-type OXTR and the products of these two mutant alleles of *OXTR* were demonstrated in in vitro assays [11]. Therefore, although there is no doubt that OXTR plays a crucial role in the proper labor, the correlation between the genetic variants of *OXTR* and preterm birth is not simple, showing a complexity of the regulation of this physiological process.

## 3. OXTR in the Regulation and Dysregulation of the Maternal/Parental Behavior

As mentioned in Section 1, OXT and OXTR were connected early to the regulation of maternal behavior, as concluded from the results of experiments with animal models, especially genetically modified mice [12]. Indeed, studies with patients indicated a connection between OXTR functions and responses to chronic stress conditions, including those occurring in the postnatal period [13]. Then, other studies conducted with humans and study subjects confirmed that there is a connection between OXTR and parental behavior. The rs2254298 single nucleotide polymorphism (SNP) in the *OXTR* gene has been associated with maternal and paternal affection towards their child [14]. The effects of the rs2254298 variants on physically controlling parenting were also confirmed [15]. Associations between rs53576 and rs2254298 polymorphisms in *OXTR* and maternal care were studied to conclude that the specific *OXTR* genotype (rs2254298 AG/AA) may be correlated with poor maternal care [16]. Another study suggested the influence of the rs968389 polymorphism on maternal sensitivity during free play with the infant [17]. On the other hand, such correlations were questioned, based on genetic studies with biological material derived from persons from different geographical and ethnic origins (European, American, and African-American) [18].

Interestingly, another correlation of the *OXTR* gene with maternal behavior has been detected when DNA methylation was studied. Namely, hypermethylation of this gene was associated with perinatal depression [19]. On the other hand, the occurrence of depression in pregnancy did not associate with DNA methylation changes at the *OXTR* locus in the cells of the placenta; however, cord plasma antidepressant levels were associated with an increased level of the methylation at the *OXTR* promotor region [20]. DNA methylation likely affects specific gene expression, and it was reported that the levels of mRNA derived from the *OXTR* gene were higher in the blood cells of mothers of infants than in no-infant women. Moreover, *OXTR* mRNA levels were lower in mothers with early trauma and less maternal experience than in the group of mothers who had not experienced trauma [21]. Intriguingly, some analyses led researchers to suggest that the “structuring behavior may buffer the potential negative impact of hypermethylation on OXTR gene expression and function” [22].

Recent investigations came back to the problem of genetic polymorphisms in the *OXTR* and behavior. When women exposed to childhood maltreatment were tested, the results of genetic analyses suggested that the rs237895 polymorphism influenced the relationship between childhood maltreatment and maternal behavior, as these two features were associated only in studied individuals who did bear the T allele, which caused a high-level expression of the OXTR gene [23]. Moreover, another study suggested that the G allele carriers of the rs53576 polymorphism might be more susceptible to the effects of severe childhood adversity [24]. Another *OXTR* polymorphism, rs1042778, was correlated with lower behavioral sensitivity, lower engagement, higher intrusiveness, and more frequent frightened/frightening behavior in mothers [25]. More reports on the associations of *OXTR* polymorphisms with maternal behavior were published recently, suggesting the correlations of specific polymorphisms (rs53576, rs2254298, rs2268493, rs1042778, or rs13316193) with perceived maternal care [26], empathy [27], maternal overprotection [28], parental rejection [29], and sensitivity of mothers to childhood parenting [30].

In summary, there are many reports indicating the associations of the *OXTR* gene polymorphisms and maternal/parental behavior; however, little information can be provided on the molecular mechanisms of such correlations. On the other hand, some investigators questioned the direct connection between the genetic variants of *OXTR* and maternal behavior, showing that more advanced studies are required to solve this problem.

## 4. OXTR in Mental Disorders

Studies on genetic and biochemical bases of mental and personality disorders are especially difficult and complicated due to the enormous complexity of both regulatory processes occurring in the central nervous system and genetic networks involved in these interactions [31,32]. The production and secretion regulation of OXT, as well as the expression of the *OXTR* gene in different parts of the brain and in different organs, have been recently reviewed [33]. The synthesis of OXT mainly occurs in neurons located in three regions of the brain, namely the supraoptic, paraventricular, and suprachiasmatic nucleus. This hormone is released from neuronal soma, axons, and dendrites. The OXT-recognizing factor, OXTR, occurs in neurons occupying various regions of the brain, thus influencing various physiological functions of this organ. In fact, OXTR was found in the cortex, hypothalamus, pons, medulla, and cerebellum [33]. Apart from the brain, OXTR is present also in cells of other organs, including the ovaries, uterus, heart, lungs, kidneys, pancreas, gastrointestinal tract, adrenal glands, and thymus. Therefore, OXT-mediated regulations influence neuronal and somatic processes which make a network of physiological responses that, if disturbed, might induce a variety of symptoms recognized as mental and/or psycho-somatic disorders. The current state of the art in the field of physiological importance of, and regulations of the brain functions and behavior by OXTR-related processes, have been recently reviewed [34]. In fact, the physiological roles of OXT and OXTR in the brain are related to the regulation of various processes, such as neuronal excitability, network oscillatory activity, synaptic plasticity, and social recognition memory [35]. Therefore, it is not surprising that any disturbances in the functions of OXTR may result in the significant impairment of the regulatory processes occurring in the brain, and, thus, in the appearance of different symptoms. Definitely, they can modify the course of various mental disorders. On the other hand, one should remember the complexity of the brain functions and difficulties in reaching solid conclusions when studying such a complicated matter.

Apart from monogenic diseases with evident mental symptoms and changes in the personalities of patients, the connections of genetic and biochemical factors with changed behavior, mental development, and cognitive abilities are often equivocal. Moreover, there are problems with the use of animal models of such diseases since many mental features are specific for humans; thus, non-human models cannot be fully adequate in assessing the specific effects of any factors or agents. Nevertheless, by analyzing genetic polymorphisms in the genes of patients and control individuals it is sometimes possible to find correlations with symptoms of mental disorders and to build hypotheses about the possible mechanisms of specific diseases. Below, we summarize and discuss the results of studies on the potential roles of OXTR in various mental disorders.

### 4.1. Autism Spectrum Disorder

Autism spectrum disorder is defined by the National Institute of Health, USA as “a neurological and developmental disorder that affects how people interact with others, communicate, learn, and behave” (https://www.nimh.nih.gov/health/topics/autism-spectrum-disorders-asd; (accessed on 5 February 2023)). The features and causes of this disease have been deeply and comprehensively reviewed recently (as examples, see refs. [36,37]). The patients manifest problems in communication with other people and in interactions with them, develop repetitive behaviors, and reveal a restricted interest in the surrounding world. Undoubtedly, autism is a complex neurobiological disorder, and the causes include both genetic and environmental factors which influence the developing brain. In fact, there is a long list of various agents, components, and parameters of different natures that were proposed as risk factors of autism spectrum disorders [36,37].

Early evidence for the connection between OXTR and autism came from observations that significant changes in plasma oxytocin OXT levels occur in affected patients. This encouraged researchers to test four single nucleotide polymorphisms (SNPs) in the *OXTR* gene in autistic patients from China. An association between autism and two SNPs (rs2254298 and rs53576) was found, providing a basis for suggesting that impaired OXTR function might contribute to the development of the disease [38]. Importantly, the same SNPs were subsequently tested in Caucasian patients and the results supported the conclusion about the association of *OXTR* with autism [39]. The hypothesis on the involvement of OXTR dysfunction in the development of autism has been also proposed on the basis of different kinds of genetic analysis. When a combined analysis of the primary genome scan data from the Autism Genetic Resource Exchange and samples derived from Finnish patients suffering from autism was performed, *OXTR* was identified as a candidate gene responsible for autism if present in a mutant form [40]. Further support for the connection of genetic changes in *OXTR* with autism spectrum disorder came from a more robust analysis in which 18 SNPs in this gene were tested. The discovery that a five-locus haplotype block, rs237897-rs13316193-rs237889-rs2254298-rs2268494, is considerably associated with autism corroborated the proposal that specific genetic variants of *OXTR* may be risk factors for this disease [41].

An interesting observation is that some SNPs in *OXTR* appeared to be risk factors in different populations, as mentioned previously for Chinese and Caucasian patients with autism [38,39]. One such SNP is rs2254298, which was also indicated as a risk factor in the Japanese population [42]. However, intriguingly, the risk allele of rs2254298 among Japanese people was identified to be ‘A’, like in the Chinese population, but contrary to Caucasian people, where the ‘G’ allele was associated with a higher risk of autism spectrum disorder. Whether ethnic differences between the Asians and Caucasians in the linkage disequilibrium or other factors can be responsible for this discrepancy remains to be elucidated. Nevertheless, such an inconsistency raised doubts on the association of *OXTR* polymorphisms with autism risk. Therefore, 18 SNPs in this gene were analyzed in samples derived from autism patients living in Ireland, Portugal, and the United Kingdom. Perhaps surprisingly, the results did not support the hypothesis about the role of common polymorphisms in the *OXTR* gene in the development of autism spectrum disorder, at least in the Caucasian population [43]. A similar conclusion could be presented based on the results of SNP studies with samples from the Slovak population [44]. On the contrary, an analysis of 14 SNPs in *OXTR* in relation to the ratios of *N*-acetylaspartate to creatine in the right medial temporal lobe in Japanese patients with autism spectrum disorder suggested again the presence of an association between *OXTR* variants and neuronal function in the medial temporal lobe, which is affected in autism [45]. These results again pointed to possible ethnic differences between the autistic patients which affect genetic analyses. However, another study performed with Japanese patients did not support the contribution of *OXTR* polymorphisms to autism spectrum disorder susceptibility [46].

As indicated above, significant controversies appeared in the interpretation of the results of genetic polymorphism studies on the possible contribution of the *OXTR* gene variants to the development of autism. This indicated a need for more detailed studies in this area. One such focused and extensive work led to the proposal that social impairment and repetitive behaviors observed in patients with autism spectrum disorder might be associated with polymorphisms in the *OXTR* 3′UTR [47]. Another complex investigation demonstrated that the cumulative genetic variation in *OXTR* impacts the reward system connectivity in patients with autism spectrum disorder, as well as in neurotypical controls [48]. The studies conducted with a large number (over 340) of autistic patients indicated that two SNPs in *OXTR*, which were controversial in their connection to the susceptibility to autism rs53576 and rs2254298, were associated with an increased severity of social deficits [49]. On the others hand, the same SNPs were correlated with fewer social deficits in patients with attention deficit hyperactivity disorder (ADHD). Therefore, it was concluded that these SNPs are not direct risk factors for impaired social abilities [49]. On the other hand, in another study, autistic patients with GA and AA genotypes of rs237902 in the *OXTR* gene revealed more severe phenotypes than those carrying the GG genotype [50].

In the light of the contradictory conclusions drawn on the basis of results from different studies on the *OXTR* gene polymorphisms and autism spectrum disorder, a meta-analysis was performed to assess if there were any connections between the SNPs in this gene and the development of the disease. The results of studies with almost 4000 autistic patients were included in this analysis, which demonstrated significant associations between autism spectrum disorder and the following *OXTR* SNPs: rs7632287, rs237887, rs2268491, rs2254298 [51]. It was proposed that animal models should be useful in studies on the role of OXTR in autism [52]. In fact, it was shown that variations in the oxytocin system contribute to differences between individual organisms in mammalian social behaviors [52].

All the above presented and discussed results indicated that analyses of *OXTR* polymorphisms are not sufficient to conclude about a role for OXTR in the development of autism spectrum disorder [53,54]. Thus, recent studies in this field involved other aspects of the *OXTR* gene and its product. One of the processes modulating the efficiency of the expression of *OXTR* is epigenetic modification, especially DNA methylation, which can probably affect the course of autism [55]. Indeed, *OXTR* methylation was demonstrated to be associated with an increased neural response within regions of the salience in the brain and with a decreased functional coupling between these regions and attentional control networks during selective social attention [56]. Moreover, higher *OXTR* methylation levels (within intron 1) were detected in autistic patients than in neurotypical subjects [57]. Another agent that can influence *OXTR* expression is the MYC-associated zinc finger protein (MAZ), a specific transcription factor. It was demonstrated that the G allele in rs1042778 of *OXTR* is a determinant for the binding of this transcription factor [58]. Therefore, the presence of the T allele in rs1042778 may impair MAZ binding, leading to less efficient transcription of the *OXTR* gene. Moreover, significantly changed densities of OXTR in the human basal forebrain and midbrain were reported in the postmortem brain tissue from individuals with autism, relative to the controls [59]. Interestingly, the correlation of three *OXTR* SNPs (rs2254298, rs53576, rs2268491) with the brain activity localized to the right supramarginal gyrus was reported in autistic patients [60]. The effects of the *OXTR* gene polymorphisms on the brain connectivity have been also shown to be dependent on sex [61], indicating that interpretations of genetic analyses must be more careful and should include gender aspects. This conclusion has been recently corroborated by results of post-mortem studies indicating that the levels of OXTR in the brains of females with autism were lower than in autistic males and healthy individuals. Such differences in the *OXTR* gene expression were also evident in the mRNA levels [62]. Furthermore, recent analyses on the use of artificial neural networks indicated that changes in methylation levels of the *OXTR* gene were specific to females with autism spectrum disorder [63].

In summary, there are many results suggesting that disturbed levels and activities of OXTR may considerably contribute to the development of autism spectrum disorder. However, there are discrepancies in conclusions about roles of the *OXTR* gene polymorphisms as risk factors for autism. In a very recent robust synthesis of published evidence of candidate genes for autism spectrum disorder, the authors failed to determine the credibility of the evidence for *OXTR* [64]. Therefore, despite the presence of published results suggesting the contribution of *OXTR* variants to the development of autism, the question about the importance of OXTR in this disease remains still unanswered.

### 4.2. Depression

Depression is an extremely broad and complex disorder, defined as a “serious mood disorder with severe symptoms that affect how people feel, think, and handle daily activities, such as sleeping, eating, or working” (https://www.nimh.nih.gov/health/topics/depression; (accessed on 5 February 2023)). The complexity of this disease manifests in a long list of symptoms, with some patients experiencing many of them while others experience only a few. Moreover, the severity of each symptom can be different in each individual patient. Depression is broadly described in the literature with proposals of potential mechanisms and causes so we indicate only a couple of recently published comprehensive review articles as examples [65,66].

As in the case of autism spectrum disorder, studies on the role of OXTR in depression started from analyses of the *OXTR* gene polymorphisms in patients with the latter disease. An association between some *OXTR* variants with unipolar depression was demonstrated [67], but the interesting point is that this concerned rs53576 and rs2254298, the same SNPs which were suggested to play a role in the development of autism. Although, the results of these early studies indicate that if OXTR dysfunction plays a role in both autism and depression, its contribution is not specific to any of these diseases but might rather facilitate disturbances in the brain functions related generally to mental disorders. Indeed, the development of a depressive-like behavior in mice treated with a selective oxytocin receptor antagonist [68] does not necessarily mean specificity of OXTR dysfunction to depression but rather may suggest a general behavioral disturbance. The rs53576 has been further correlated with depression, as it was reported that this *OXTR* polymorphism, particularly the ‘A’ allele, may be partially responsible for the transmission of maternal depression to youth [69]. An interesting hypothesis was proposed that the mechanism by which defects in *OXTR* influence the development of depression is based on the dysfunctional social processes occurring in the absence of fully active OXTR [69]. However, we propose that the general disturbance in behavior, rather than the development of a specific disease, may be a possible indirect effect of the presence of specific *OXTR* polymorphisms. Support for such a proposal arises also from the results of studies in which correlations between the *OXTR* genotype and anxiety, stress, and depression scores were depicted [70].

The rs53576 polymorphism has been widely analyzed in subsequent studies on depression; however, the conclusions from various studies were again (as in the case of autism spectrum disorder) different. A decreased level of methylation of the *OXTR* exon 1 was identified in depressed female patients, but this association was modulated by the rs53576 polymorphism [71]. On the other hand, when *OXTR* methylation and rs53576 were investigated in another study, a greater DNA methylation was observed in patients with depression, but only in the presence of the AA genotype at rs53576 [72]. Yet another study led to presentation of the conclusion that the GG genotype at rs53576 results in greater odds of postpartum depression in women, which correlated with enhanced DNA methylation level in the *OXTR* locus, whereas methylation is unrelated to this kind of depression in the presence of the ‘A’ allele [73]. As evidenced above, the published conclusions on the role of *OXTR* polymorphism and methylation in depression are contradictory, making the question unanswered. There were further studies showing the influence of the rs53576 polymorphism on different aspects of depression, such as negative social interactions [74], prepulse inhibition of the startle reflex and startle reactivity [75], interpersonal risk factors [76], suicide attempts [77], hippocampal volume [78], effects of social environment on postpartum depression [79,80], negative affectivity [81], trauma-related psychopathology [82], and work stress [83]. However, other reports presented results showing no correlations between depression and the rs53576 polymorphism in the *OXTR* gene [84,85,86,87,88]. These discrepancies did not solve the problem, but rather deepened the confusion and ambiguity about the putative role of the *OXTR* polymorphism in depression.

There were attempts to correlate the level of methylation of the *OXTR* gene with the severity of depression, however, no significant association between these two parameters could be found, despite quite a large number (846) of tested patients [89]. On the other hand, earlier studies on *OXTR* methylation strongly suggested that functions of the oxytocin system, including OXTR-mediated signal transduction, may be involved in the attenuation of the fear response, which can protect against depression [90]. Thus, we suggest that disturbances in the expression of the *OXTR* gene, rather than the gene polymorphisms per se, might facilitate the development of this disease. Such a hypothesis can be supported by the results of studies on chronic stress, which is both the major risk factor for depression and significantly influences *OXTR* expression [90]. Indeed, earlier work suggested that the rs53576 variants confer vulnerability for depression within the context of interpersonal risk factors [91], while a very recent report demonstrated that the ‘A’ variant of rs53576 results in upregulation of the *OXTR* gene expression, though independently from the DNA methylation status [92].

The above presented conclusion and hypothesis on the crucial role of the level of expression of the *OXTR* gene and resultant abundance of the receptor may also have implications for understanding the roles of OXTR in other diseases. These will be discussed in the next sections of this paper.

### 4.3. Schizophrenia

Schizophrenia is defined as “a serious mental illness that affects how a person thinks, feels, and behaves” (https://www.nimh.nih.gov/health/publications/schizophrenia; (assessed on 5 February 2023)). Among the numerous symptoms of this disease, one can distinguish psychotic (hallucinations, delusions, thought disorder, and movement disorder), negative (trouble with planning and sticking with activities, trouble with anticipating and feeling pleasure, talking in a dull voice, avoiding social interaction, severely decreased life energy), and cognitive (trouble with processing information and making decisions, inability to immediately use information, inability to focus or pay attention) ones. The description of this disease is broad, and recent review articles provide a great background for its causes, mechanisms (which are still only partially understood), and treatment possibilities [93,94,95,96].

When the role of OXTR in schizophrenia was tested, it was demonstrated that in the brains (particularly in the temporal cortex) of patients (tested post-mortem), the levels of mRNA of the *OXTR* gene were significantly decreased relative to the control samples. Moreover, a decrease in the OXTR binding was found in the vermis [97]. On the other hand, higher *OXTR* mRNA levels were detected in leukocytes of first-episode schizophrenia patients than in healthy persons [98]. The enhanced expression of the *OXTR* gene, at mRNA and protein levels, in the blood cells derived from the patients has been recently confirmed [99]. Therefore, the expression of the *OXTR* gene may be different in the brains and in the peripheral tissues of schizophrenia patients; however, dysregulation of this gene in this disease is evident. Interestingly, when post-mortem studies with the brains of patients suffering from major depressive disorder, bipolar disorder, and schizophrenia were tested, the levels of *OXTR* mRNA were increased in the dorsolateral prefrontal cortex [100]. Again, it appears, therefore, that the mechanisms of the involvement of dysfunctions of OXTR in major mental disorders might be common, or at least similar.

Epigenetic changes in the *OXTR* gene were also reported as being associated with schizophrenia. Namely, significantly decreased *OXTR* methylation was reported in cells from the peripheral whole blood in the patients relative to the controls [101]. The presence of the rs53576 polymorphism was associated with disturbed social cognition abilities in schizophrenia patients [102], which, in combination with the previously mentioned changes in expression of the *OXTR* gene [92], can corroborate the suggestions that OXTR deficiency may cause social behavior-related defects in various mental diseases.

### 4.4. Other Mental Disorders

Connections of various dysfunctions of OXTR and the oxytocin system with not only autism, depression, and schizophrenia, but also different mental disorders were reported. Here, we will mention studies which might shed further light on the mechanisms by which OXTR may influence the mental development.

As in the mental disorders discussed above, the role of OXTR in obsessive-compulsive disorder is also controversial. Some results indicated no significant associations between any of several tested SNPs in the *OXTR* gene [103], while other studies led to the conclusion that such SNPs can modulate the onset age of this disease, thus, playing an important role in the pathophysiology [104]. Another discrepancy appeared during studies on DNA methylation in the *OXTR* locus, as there are reports demonstrating enhanced DNA modification in the cells of patients with obsessive-compulsive disorder [105,106] while other analyses led to the opposite conclusions, pointing to the impaired methylation of *OXTR* [107].

Interestingly, recent investigations of *OXTR* methylation in patients suffering from various mental disorders revealed that the level of this modification varies significantly between patients; however, those with extreme levels had lower intelligence quotient (IQ) scores and experienced more social problems than the patients with the methylation efficiency comparable to that in the healthy controls [108]. Once more, these results corroborate the proposal that changes in OXTR are not specific to any individual mental disorder but rather that they can modulate the course of different diseases by influencing social behaviors and/or cognitive functions. This can be further supported by reports indicating that various polymorphisms in the *OXTR* gene can influence the antisocial behavior in adolescent boys [109], attention deficit/hyperactivity disorder [110], and aggressive behaviors [111,112].

An especially interesting and inspiring report has been published recently in which human post-mortem brain samples were analyzed for the efficiency of expression of the *OXTR* gene [113]. Brains derived from persons who died at very different stages of development, from the prenatal period to late adulthood, were investigated. Intriguingly, the expression of the *OXTR* gene was found to be increasing during the prenatal period, while the highest levels of the expression were detected in early childhood. A comprehensive analysis revealed an enrichment in a network of the expression of genes functionally coupled with *OXTR* in several mental disorders [113]. That work strongly corroborated the important role of OXT and OXTR in crucial processes related to mental development.

## 5. OXTR in Mucopolysaccharidoses

Mucopolysaccharidoses (MPS) are inherited metabolic disorders belonging to lysosomal storage diseases (LSD), which are characterized by the accumulation of partially degraded glycosaminoglycans (GAGs) [114,115]. Depending on the kind of deficient enzyme involved in the degradation of GAGs (due to mutations in the corresponding genes) and the nature of the accumulated GAG(s), 13 types and subtypes of MPS are currently distinguished [116]. All MPS types are severe diseases, and neurodegenerative processes, accompanied with mental deficits and disorders, occur in most of them [114,116,117]. Although MPS are monogenic diseases, recent studies indicated that the expression of hundreds of genes is changed (either up- or down-regulated) in each MPS type relative to the controls [118]. This causes a battery of secondary cellular changes which contribute significantly to the deterioration of the functions of cells, tissues, organs, and, finally, the whole organisms, some of which can be hardly reversible or even irreversible [115,119,120]. Unfortunately, despite enzyme replacement therapy being currently available for a few MPS types, it can improve only some disease symptoms, while those related to the brain functions remain largely untreatable. Thus, patients suffering from neuronopathic forms of MPS still lack specific treatments that might improve their functioning and restrict behavioral problems and cognitive deficits [121].

Interestingly, it appears that the primary GAG storage is only the trigger of subsequent devastating changes, rather than the main or the only cause of the disease [122]. For example, the formation of protein aggregates (such as amyloid depositions) and autophagy dysfunction were reported as pathological processes downstream of the GAG storage, which may severely impair cellular functions and cause further changes in the structures and/or activities of organelles and efficiencies of biochemical processes [123,124]. Indeed, different disturbances in cell physiology were reported in various MPS types, which significantly contribute to the disease severity [125,126,127].

Global transcriptomic analyses revealed that the expressions of many genes related to human behavior were dysregulated in all types of MPS [128]. Among them, the *OXTR* gene was found to be one of the most affected, indicating a significant up-regulation (between 3- and 13-fold, depending on the MPS type), as estimated by both RNA-seq and RT-qPCR analyses [128]. This was an interesting discovery in the light of MPS symptoms, which include, but are not restricted to, severe behavioral and social problems, such as aggressive-like behavior, hyperactivity, attention deficit, and mental retardation [114,129]. Such symptoms resemble those described in the preceding section (Section 4) as characteristic of disorders associated with OXTR changes. Indeed, the symptoms of some MPS types, especially all subtypes of MPS III (Sanfilippo disease) are so similar to autism spectrum disorder or attention deficit hyperactivity disorder that MPS III is often misdiagnosed as one of these diseases [130,131,132,133].

The results of recent molecular studies provided a possible explanation for the OXTR-related modulation of the pathomechanism of MPS. Namely, it was found that OXTR can directly interact with GAGs (stored in MPS cells), forming large aggregates [134]. Such aggregated forms of OXTR are inactive; thus, the functions of this receptor are impaired. Therefore, even in the presence of higher levels of OXTR (as demonstrated by biochemical analyses [134]), there is a deficit of active OXTR molecules which may contribute to the development of specific behavioral symptoms in MPS patients. The role of GAGs in the modulation of *OXTR* expression and the formation of the OXTR-containing aggregates was confirmed in experiments with the agents causing a decrease in GAG storage, like supplementation with either an enzyme allowing efficient degradation of these compounds or a compound impairing their synthesis. In both cases, the levels of *OXTR* mRNA and OXTR protein, as well as the abundance of OXTR aggregates, decreased significantly [134].

The studies described above led to two important conclusions. First, the problems with OXTR in MPS, and perhaps also more generally in some other diseases, might appear not only due to the presence of specific SNPs in *OXTR* and/or changes in DNA methylation, leading to the dysregulated expression of the *OXTR* gene, but also because of the interactions of this receptor with other compounds and formation of inactive aggregates, lowering its actual activity. Second, the deposition of protein complexes in MPS cells, which impairs the autophagy process and leads to further cellular dysfunctions, concerns not only amyloids (as reported previously [123,124], but also other proteins (exemplified by OXTR [134]) that are otherwise important factors in regulating homeostasis. Therefore, investigations of OXTR malfunctions provided important data which facilitate the understanding of both detailed pathomechanisms of specific diseases, such as MPS, and general dysregulation of homeostasis due to changes in the activity of this receptor.

## 6. OXTR in Cancer

The involvement of OXTR in cancer development was suggested several times. SNPs in the *OXTR* gene were associated with an increased risk of Barrett’s esophagus, a premalignant condition, and esophageal adenocarcinoma [135]. Alternations in the *OXTR* gene were found in patients with hepatocellular carcinoma; however, these changes, though statistically significant in the association analyses, occurred in only 3% of affected individuals [136].

Increased levels of *OXTR* mRNA were correlated with the occurrence of other cancers. Namely, the expression of this gene was up to 10-fold up-regulated in pancreatic cancer cells [137], and increased levels of the corresponding mRNA were noted in colorectal cancer [138]. Moreover, OXTR has been proposed to play a role in breast cancer development and progression, though mechanism(s) of this connection remain(s) to be elucidated [139]. On the other hand, the down-regulation of the *OXTR* gene expression was reported recently in breast cancer relative to the non-cancer tissue [140,141]. In contrast, mammary tumorigenesis was induced by the overexpression of *OXTR* in a mouse model [142].

Interestingly, the use of data related to *OXTR* was proposed for prognostic purposes in different cancers. Namely, *OXTR*-derived mRNA levels were significantly increased in malignant mesothelioma cells, and the higher expression efficiency correlated with the poor prognosis [143]. The *OXTR* was among genes whose changed expressions were classified as potential prognostic markers of lower-grade glioma [144,145]. Then, the elevated levels of *OXTR* mRNA were suggested as an indicator of the poor prognosis of colon adenocarcinoma [146] and colorectal cancer [147]. Finally, the increased efficiency of *OXTR* expression was also characteristic for oral squamous cell carcinoma [148].

Generally, the results of studies on the involvement of OXTR changes in cancer corroborate the conclusions presented in Section 4 and Section 5 (‘OXTR in Mental Disorders’ and ‘OXTR in Mucopolysaccharidoses’) that the efficiency of the expression of the *OXTR* gene rather than SNPs themselves result in the effective modulation of the disease course. Again, this appears to be a common feature of many diseases, rather than being restricted to individual disorder(s).

## 7. OXTR in Cardiovascular Diseases

No significant associations were found between the *OXTR* gene polymorphism and cardiovascular risk factors [149]. Such a conclusion on the lack of considerable correlations between *OXTR* genotype and cardiovascular disease was corroborated in studies, indicating that the rs2268498 SNP in the *OXTR* gene is not a risk factor for hypertension [150]. Nevertheless, increased levels of expression of this gene have been reported in humans as well as animal models of atherosclerosis [151], vascular dementia [152], and cardiomyopathy [153].

A comprehensive review of OXTR in vascular functions and stroke has been published recently [154], where the authors analyzed the signal transduction processes mediated by this receptor protein in the functionality of the vascular system (thus, we will not discuss the details here again). Such analyses allowed them to suggest possible mechanisms of the involvement of OXTR in the development of cardiovascular diseases [154]. At the molecular level, it seems that the OXTR initiated (after binding of OXT) signal transduction, leading to the enhanced expression of the gene coding for Bcl-2, which has a pro-survival function and might be the crucial process.

In another recent comprehensive review on the role of OXT and OXTR in cardiovascular diseases [33], it was stressed that the secretion of oxytocin is modulated in various conditions, including hypertension and myocardial infarction. This, in turn, affects the expression of the *OXTR* gene. Therefore, it was concluded that the pathogenesis of cardiovascular diseases can be significantly influenced by the dysregulation of OXT production and *OXTR* expression, especially in such disorders as ischemia, hypoxia, inflammatory disturbances, pain, and stress conditions [33]. Because the mechanisms of these processes were carefully discussed in the review article mentioned above, we are not repeating this to avoid redundancy and, instead, refer readers to that article [33].

## 8. OXTR in Other Diseases

Other diseases might also be modulated by the functions and dysfunctions of OXTR. For example, the signal transduction pathway initiated by this receptor regulates the osteoblast/adipocyte balance [154]. Studies conducted with the mouse models suggest that the deficiency in this process might contribute to osteoporosis, while the stimulation of this pathway by OTX could potentially reverse this disease [155]. OXTR functions can also affect obesity. Higher frequencies of ‘GG’ and ‘AG’ genotypes at the rs53576 polymorphism of the *OXTR* gene were reported in obese persons relative to control ones [156]. The efficiency of *OXTR* methylation was significantly lower in the obese group with binge eating disorder; however, this correlation occurred only in males, not in females [157]. It is also worth mentioning that the expression of the *OXTR* gene, and thus the abundance of OXTR, can be significantly modulated by external factors, such as viral infections [158] and inflammatory processes [159]. Therefore, such conditions may considerably influence all processes controlled by OXTR functions, including the course of various diseases in which this receptor plays roles. Such diseases can be modulated in a double way, by infection/inflammation themselves(s) and through OXTR-mediated signal transduction modification.

Interestingly, apart from the effects of OXTR dysfunction on the pathomechanism of mucopolysaccharidoses (described in Section 5), the role of this receptor in the course of another genetic disorder, Williams syndrome, has been recently evaluated [160]. This syndrome is caused by a deletion (encompassing some 25 genes) in the q11.23 region of the chromosome 7. It is a neurodevelopmental disorder, and the patients manifest cognitive deficits, behavioral disorders, emotional problems, and social profile disturbances [161]. The inspiration for the above-mentioned analysis was the discovery that the expression of the *OXTR* gene was impaired and the *OXTR* gene region was hypermethylated in the blood cells of patients suffering from Williams syndrome relative to healthy controls [162]. Therefore, an analysis of the available data on the efficiency of expressions of genes coding for OXT and OXTR and their correlations with the investigated disease has been conducted [160]. The authors of that report predicted that there is an epigenetic control (based on DNA methylation) of the social behavior, as well as the influence of SNPs in the *OXTR* gene on the development of symptoms of Williams syndrome. They suggested that a better understanding of the role of OXTR in this disease should facilitate the development of an efficient treatment for patients. Together with the studies discussed in Section 5, these analyses indicated that the changes in the levels and/or activities of OXTR may contribute to the pathomechanisms of genetic disorders, not as primary causes, but rather as important modulators of cellular dysfunctions and modifiers of the course of diseases.

## 9. Concluding Remarks

As summarized in this review, the changes related to OXTR significantly influenced the course of various diseases, from reproductive system diseases and through mental disorders, mucopolysaccharidoses, cancer, and cardiovascular diseases to others (such as osteoporosis and obesity). They are presented schematically in Figure 1.

However, the molecular mechanisms by which OXTR affects these diseases are mostly unknown. Our knowledge in this matter is based predominantly on the associations between various polymorphisms of the *OXTR* gene, methylation efficiency of this gene, and/or changes in the levels of *OXTR* mRNA with occurrence of selected symptoms or diagnosed disorders. The picture is even more complicated because exactly the same SNPs have been reported to affect different diseases. Nevertheless, this fact led us to propose the hypothesis that the effects of OXTR are not specific to individual diseases but rather that they can affect common processes which might, in turn, modulate the course of different disorders. Intriguingly, there are many examples of contradictory results published by different authors which demonstrated either significant associations of specific *OXTR* SNPs with particular diseases or a complete lack of correlations in studies on the same SNPs in the same diseases. Such confusions concern also *OXTR* methylation. Moreover, considerable differences were reported between sexes and between different tissues. However, we propose that the ostensible paradox of contradictory results obtained by various research groups might be explained in such a way that similar variabilities in *OXTR* polymorphisms or DNA methylation may occur in both patients suffering from different diseases (especially mental disorders) and healthy controls; however, the effects of specific SNPs or levels of methylation can be significantly more pronounced in affected persons due to their influence on ongoing pathological processes and enhancement of symptoms. The same disturbances in OXTR-mediated regulations might be masked in healthy persons due to fully functional other control processes. This can be especially pronounced in the effective controlling of emotions and behaviors which are otherwise disturbed in people with mental disorders. Therefore, depending on the compositions of investigated groups (note that the tested cohorts consisted of relatively low number of persons, such as several dozen or a few hundred), the manifestations of (sometimes subtle) differences in symptoms related to OXTR functions might be more or less pronounced (or, in other words, the masking of mental disturbances might be less or more effective) and significantly affect the results of statistical analyses of the influence of the *OXTR* status on the investigated disorders.

We also emphasize that the efficiency of expression of the *OXTR* gene, rather than the presence of its specific polymorphic variant, may considerably influence the signal transduction process mediated by OXTR, and then, all cellular and further organismal changes related to the activities of a battery of genes controlled by this molecular signaling. Such a scenario is compatible with the results of mechanistic investigations which are, unfortunately, still relatively scarce. As presented in this review, one of few molecular mechanisms of the effects of OXTR malfunctions in human diseases was reported in studies on mucopolysaccharidoses. In this case, elevated levels of OXTR, resulting from enhanced transcription of the *OXTR* gene, do not correspond with its higher activity, due to the direct interactions with accumulated glycosaminoglycans (stored in large amounts in this disease), which cause the formation of aggregates, leading to the inactivation of the receptor and the impairment of the signal transduction process. Definitely, an understanding of the molecular details of the mechanisms of OXTR-dependent changes in other diseases is necessary to fully assess the affected processes and to enable us to predict the effects of various changes in OXTR on the development of specific symptoms of different diseases. Then, the development of effective therapies targeting OXTR may be potentially efficient in the future.

## Figures and Tables

**Figure 1 ijms-24-03887-f001:**
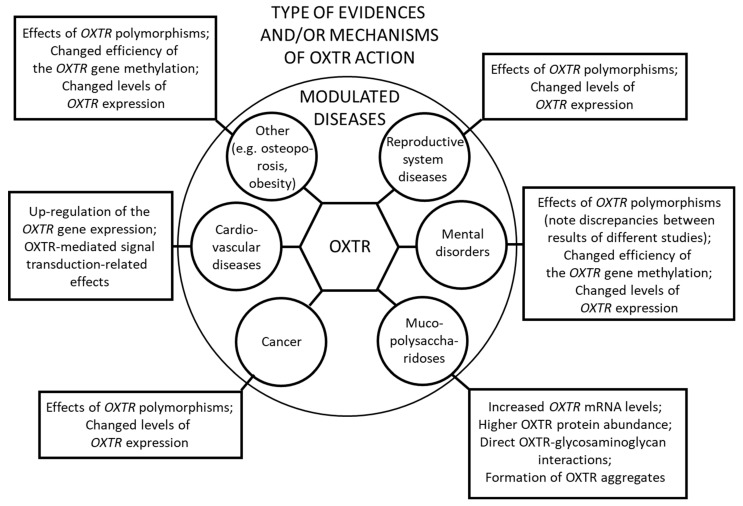
A simplified overview of the effects of OXTR on human diseases. Target (modulated) groups of diseases are indicated, together with major types of evidence of the influence of OXTR and/or mechanisms of actions of this receptor in specific diseases.

## Data Availability

Not applicable.

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
