# Peer review of "Roles of the Oxytocin Receptor (OXTR) in Human Diseases"

_ijms, 2023, doi:10.3390/ijms24043887_

Round 1

Reviewer 1 Report

This review presents many negative or ambiguous and inconclusive results with some positive ones. However, this is not the authors' fault but a reflection of the topic's complexity. Different papers present different results, often inconsistent, and sometimes contradictory. This review is worth publishing since at least a partial ordering of knowledge on this important issue is much needed. The strength of the article is the independent creation of its hypothesis about the importance of OXTR gene variation in the pathogenesis of various diseases. Fig. 1 is very helpful.

Some detail issues:

1. p.2, 63; rather than  "gynecological disorders" apply "reproductive system diseases"; gynecology is a medical specialty only

2. p. 2, 95; cleared showed - please correct

3. p. 3, 101; should be "changed"

4. p. 5, 206; acronym MAZ requires an explanation

5. p. 8, 376; should be "OXTR gene"

Reviewer 2 Report

Article review

Title: Roles of the oxytocin receptor (OXTR) in human diseases

Authors: Karolina Pierzynowska, Lidia Gaffke, Magdalena Żabińska, Zuzanna Cyske , Estera Rintz, Karolina Wiśniewska, Magdalena Podlacha, Grzegorz Węgrzyn

Journal: International Journal of Molecular Sciences (IF 6.208)

Reviewer comment to the manuscript:

Authors of the manuscript “Roles of the oxytocin receptor (OXTR) in human diseases” are presenting wholesome review of the impact of oxytocin receptor on various aspects of human pathology with focus on Gynecological Disorders; Mental disorders; Mucopolysaccharidoses; Oncology field (chapter titled as OXTR in cancer); Cardiovascular disease; followed by paragraph titled “OXTR in Other Disease”. Whole main text of the manuscript is followed with Conclusion remarks paragraph (chapter 7 of the main manuscript).  Manuscript contain one self-explanatory figure with very simplified view of all described connection between OXTR and its involvement in human diseases. Authors declare 106 references, the oldest one dated to 2005.

Main idea and goals related to reviewed manuscript are understandable and well welcomed by the science community, unfortunately overall impact is more or less missed. Whole main text of the reviewed manuscript is highly imbalanced. Manuscript is highly focused on connections between OXTR and Brain related pathology (chapter 3 OXTR in Mental disorders). Unfortunately, other chapters are seriously underrated or overlooked in mater of bringing new discoveries and recent scientific works to the targeted readers. Main focus to OXTR and its connection to the Single Nucleotide Polymorphism, DNA methylation, pathology, pathophysiology and diseases as whole picture is understandable, however physiological function of the OXTR, its interaction with Oxytocin and / or Oxytocin impact on any mentioned disease or syndrome is mentioned superficially or not mentioned at all. We, targeted readers, can eventually reach conclusion that OXTR is main molecular agent; and its pathophysiological impact is maintained without its main ligand oxytocin. Molecular aspect of the OXTR as a G-Coupled Receptor is not mentioned at all.

Reviewer comments and questions:

1.       I would highly recommended to include molecular mechanisms of OXTR and its impact to pathophysiological processes.

2.       Third sentence in the introduction (row 37) is declaring that OXTR and oxytocin itself is “are primarily known for controlling maternal behavior” but related chapter is not included with the reviewed manuscript. Please include and elaborate this topic.

3.       In the row 38 authors declare “OXTR is mainly formed in the myoepithelial cells of the mammary gland and in the myometrium and endometrium of the uterus at the end of pregnancy.” Please elaborate this statement and / or confront this statement with the newest scientific knowledge.

4.       In the chapter 3 (OXTR in Mental Disorders), especially subchapter 3.1 Autism spectrum disorder, I would highly recommend to include more information related to OXTR and Oxytocin impact on Neuronal precursor cells, Neurons and glial cells; impact in different brain areas; impact during prenatal and postnatal period of Central Nervous System. Without focus on these areas can be eventually whole subchapter 3.1 Autism spectrum disorder considered as insufficient at least.

5.       Chapter related to Depression (3.2) and Schizophrenia (3.3) should be elaborate as well, because information mentioned in this area are insufficient. I would highly recommend to seriously discriminate between all units and aspects related to “Depression and Schizophrenia”, including cause of depression and disturbances in neurotypical brain physiology.

6.       Chapter related to Other mental disorders (3.4) is mentioning “various mental disorders” have to be widely elaborated.

7.       Chapter 4. OXTR in Mucopolysaccharidoses is well described and focused on topics, however whole chapter is backed by references related to (or at least looks like) almost the same people and / or research group. There are no other out-of-the-box references. Is this chapter written as a significant research discovery in the field? If yes, it should be elaborated and mentioned in the Discussion section. Please consider to include at least internal discussion in the chapter.

8.       Chapter 5. OXTR in Cancer; chapter 6. OXTR in Cardiovascular Diseases; and chapter 7. OXTR in Other Diseases should be widely elaborated and I would recommend to include more recent information related to the field. Otherwise these chapters are seriously overlooked and making whole manuscript look imbalanced.

Thank you for your cooperation.

Round 2

Reviewer 2 Report

No further comments to be addressed at this point.